# Effect of Freeze–Thaw Cycling on the Failure of Fibre-Cement Boards, Assessed Using Acoustic Emission Method and Artificial Neural Network

**DOI:** 10.3390/ma12132181

**Published:** 2019-07-07

**Authors:** Tomasz Gorzelańczyk, Krzysztof Schabowicz

**Affiliations:** Faculty of Civil Engineering, Wrocław University of Science and Technology, Wybrzeże Wyspiańskiego 27, 50-370 Wrocław, Poland

**Keywords:** fibre-cement boards, non-destructive testing, acoustic emission, artificial neural networks, SEM

## Abstract

This paper presents the results of investigations into the effect of freeze–thaw cycling on the failure of fibre-cement boards and on the changes taking place in their structure. Fibre-cement board specimens were subjected to one and ten freeze–thaw cycles and then investigated under three-point bending by means of the acoustic emission method. An artificial neural network was employed to analyse the results yielded by the acoustic emission method. The investigations conclusively proved that freeze–thaw cycling had an effect on the failure of fibre-cement boards, as indicated mainly by the fall in the number of acoustic emission (AE) events recognized as accompanying the breaking of fibres during the three-point bending of the specimens. SEM examinations were carried out to gain better insight into the changes taking place in the structure of the tested boards. Interesting results with significance for building practice were obtained.

## 1. Introduction

Fibre-cement boards have been used in construction since the beginning of the last century. Their inventor was the Czech engineer Ludwik Hatschek, who developed and patented the technology of producing fibre cement, then called “Eternit”. The material was strong, durable, lightweight, moisture-resistant, freeze–thaw resistant and non-combustible [1]. Fibre cement became one of the most popular roofing materials in the world in the 20th century. This was so until one of its components (i.e., asbestos) was found to be carcinogenic. In the 1990s, asbestos was replaced with environment-friendly fibres, mainly cellulose fibres. The fibre-cement boards produced today are made up of cement, cellulose fibres, synthetic fibres and various additives and admixtures. They are a completely different building product than the original one [2], and still require investigation and improvement. The additional components and fillers of fibre-cement boards are lime powder, mica, perlite, kaolin, microspheres and recycled materials [3,4], whereby fibre-cement boards can be regarded as an innovative product which fits into the sustainable development strategy. At present, such boards are used in construction mainly as ventilated façade cladding [5], as illustrated in Figure 1. In the course of their service life, fibre-cement boards are exposed to various factors, such as chemical (acid rains) and physical aggressiveness (ultraviolet radiation), but mainly to variable environmental impacts, including sub-zero temperatures in the winter season.

After a few winter seasons, the effect of sub-zero temperatures—especially of temperature (freeze–thaw) cyclicity—needs to be determined in order to establish whether the fibre-cement boards can remain in service as ventilated façade cladding. The knowledge of this effect is essential not only from the scientific point of view, but also for building practice. It is worth noting that research on fibre-cement boards has so far been mostly limited to determining—solely through the bending strength (modulus of rupture, *MOR*) test—their standard physicomechanical parameters and the effect of in-service factors (e.g., soaking–drying cycles, heating–raining cycles and high temperatures) and the various fibres and production processes [6]. Only a few cases of testing fibre-cement boards by non-destructive methods, limited to imperfections arising during the production process, can be found in the literature [7,8,9,10]. Besides the effects of high temperature and fire described in [11,12], the impact of sub-zero temperatures is one of the most destructive in-service factors to many building products, particularly composite products containing reinforcement in the form of various fibres, especially fibres of organic origin (to which cellulose fibres belong). In the authors’ opinion, freeze–thaw cycles can very adversely affect the durability of such composites. Experiments were carried out in order to prove this thesis. The experiments consisted of subjecting fibre-cement board specimens to 1 and 10 freeze–thaw cycles and then investigating them under three-point bending by means of the acoustic emission (AE) method. Artificial intelligence in the form of an artificial neural network [13] was employed to analyse the experimental results. Previous studies by the authors [11,12,14,15] presented the assessment of the effect of freeze–thaw cycling based solely on bending strength to be inadequate. Whereas in this study, using the acoustic emission technique and analysing the degradation of the specimens, the authors were able to describe the degrading changes in the structure of the tested boards on the basis of not only the mechanical parameters, but also the acoustic phenomena. The registered AE signals provided the basis for developing reference acoustic spectrum characteristics accompanying cement matrix cracking and fibre breaking during bending. Then, an artificial neural network was used to recognize the characteristics in the AE records. In the course of freeze–thaw cycling, the fibres in the boards gradually degraded, which manifested as a fall in the number of events recognized as accompanying the breaking of fibres. This is described in more detail later in this paper. In order to verify the results and gain better insight into the changes taking place in the structure of the fibre-cement boards, they were examined under a scanning electron microscope (SEM).

## 2. Literature Survey

To-date, research on fibre-cement boards has focused on the effect of in-service factors [16,17,18] and the effect of high temperatures, determined by testing the physicomechanical parameters of the boards—mainly their bending strength (*MOR*). Only a few cases of testing fibre-cement boards by non-destructive methods, including the acoustic emission method, have been reported in the literature. Ardanuy et al. [6] presented the results of investigations into the effect of high temperatures on fibre-cement boards, but were limited to the bending strength test. Li et al. [19] examined the effect of high temperatures on composites produced using the extrusion method, but solely on the basis of the mechanical properties of the composite. Schabowicz et al. [11,12] used non-destructive methods to assess the effects of high temperature and fire on the degree of degradation of fibre-cement boards on the basis of the physicomechanical parameters. Other reported investigations of fibre-cement boards were devoted to the detection of imperfections arising during the production process. Papers by Drelich et al. [8] and Schabowicz et al. [20] presented the possibility of exploiting Lamb waves in a non-contact ultrasound scanner to detect defects in fibre-cement boards at the production stage. A method of detecting delaminations in composite elements by means of an ultrasonic probe was presented in a study by Stark, Vistap et al. [21]. Ultrasonic devices and a method used to detect delaminations in fibre-cement boards were described by Dębowski et al. [7]. Berkowski et al. [22], Hoła and Schabowicz [23] and Davis et al. [24] proposed the use of the impact-echo method jointly with the impulse response method to recognize delaminations in concrete elements. However, it is not recommended to test fibre-cement boards in this way since the two methods are intended for testing elements which are thicker than 100 mm. The special hammer used in the impulse response method can damage the fibre-cement boards being tested, while in the impact-echo method multiple wave reflections cause disturbances which make it difficult to interpret the obtained image [22]. Therefore, it is inadvisable to use the two methods to test fibre-cement boards, which are about 8 mm thick. There is scant information in the literature on the use of other non-destructive methods to test fibre-cement boards. The preliminary research described in [9,25] showed the terahertz (T-Ray) method to be suitable for testing fibre-cement boards. The character of terahertz signals is very similar to that of ultrasonic signals, but their interpretation is more complicated. Schabowicz et al. [15] and Ranachowski et al. [26] used X-ray microtomography to identify delaminations and low-density regions in fibre-cement boards. This technique was found to precisely reveal differences in the microstructure of such boards. It can be a useful tool for testing the structure of fibre-cement boards in which defects can arise as a result of production errors, but it is applicable only to small boards. As already mentioned, only a few cases of testing fibre-cement boards by means of acoustic emission have been reported so far. Ranachowski et al. [26] carried out pilot tests on fibre-cement boards produced by extrusion, including boards exposed to the temperature of 230 °C for 2 h, using the acoustic emission method to determine the effect of cellulose fibres on the strength of the fibre-cement boards and tried to distinguish the AE events emitted by the fibres from the ones emitted by the cement matrix. The investigations showed this method to be suitable for testing fibre-cement boards. Schabowicz et al. [11,12] and Gorzelańczyk et al. [11] proposed the use of the acoustic emission method to study the effects of fire and high temperatures on fibre-cement boards. The effect of high temperatures on concrete has been studied using the acoustic method (e.g., by Ranachowski [27] and Ranachowski et al. [28,29,30]), and is described widely in the literature. A large quantity of data are recorded during acoustic emission measurements, and they need to be properly analysed and interpreted. For this purpose, it can be useful to combine the acoustic emission method and artificial intelligence, including artificial neural networks (ANNs). ANNs are used to analyse and recognize signals acquired during the failure of various materials [31]. In [32,33,34] ANNs were used to analyse the results of testing concrete by means of non-destructive methods. Łazarska et al. [35] and Woźniak et al. [36] in their investigations of steel successfully used the acoustic emission method and artificial neural networks to analyse the obtained results. Rucka and Wilde [37,38], Zielińska and Rucka [39] and Wojtczak and Rucka [40] successfully used the ultrasonic method to investigate damage to concrete structures and masonry pillars. ANNs were also successfully used by Schabowicz et al. [11,12] to analyse the results of tests consisting of exposing fibre-cement boards to fire and high temperature.

Considering the above information, the authors came to the conclusion that the acoustic emission method combined with artificial neural networks would be suitable for assessing the changes taking place in the structure of fibre-cement boards exposed to freeze–thaw cycling.

## 3. Strength Tests

Two series of fibre-cement boards, denoted respectively A and B, were tested to determine the effect of freeze–thaw cycling. Altogether 60 specimens were tested. The basic specifications of the boards in the two series are given in Table 1.

The freeze–thaw cycling of the specimens was conducted as follows. First the specimens were cooled (frozen) at a temperature of −20 ± 4 °C in a freezer for 1–2 h and kept at this temperature for the next hour. The specimens were heated (thawed) in a water bath at a temperature of 20 ± 4 °C for 1–2 h and kept at this temperature for the next hour. One should note that the above temperatures apply to the conducting medium (i.e., air or water). Each freeze–thaw cycle lasted on average about 5–6 h. Air-dry reference specimens (not subjected to freeze–thaw cycling) were denoted as A_R_ and B_R_. The denotations of exemplary series of boards are presented in Table 2. Figure 2 shows exemplary views of the tested (20 × 100 mm and 8 mm thick) specimens.

In order to determine the effect of freeze–thaw cycling on the fibre-cement boards, the latter were subjected to three-point bending and investigated using the acoustic emission method. Breaking force *F*, strain *ε* and AE signals were registered in the course of the three-point bending. Figure 3 shows the three-point bending test stand and the acoustic emission measuring equipment.

The curve of flexural stress *σ*_m_, the flexural strength (*MOR*), the limit of proportionality (*LOP*) and strain *ε* were taken into account in the analysis of the experimental results. The *MOR* was calculated from the standard formula [41]:(1)$$MOR=\frac{3Fl_{s}\;}{2b\;e^{2}},$$
where:
*F* is the loading force (N);*l_s_* is the length of the support span (mm);*b* is the specimen width (mm); and*e* is the specimen thickness (mm).

Figure 4 shows *σ*–*ε* graphs under bending for the specimens of all the tested fibre-cement boards.

Figure 4 shows that as a result of freeze–thaw cycling, flexural strength (*MOR*) decreased by 35%–50% in comparison with the flexural strength (*MOR*) of the reference fibre-cement boards. No significant difference in the change of bending strength (*MOR*) between the specimens subjected to one or ten freeze–thaw cycles was noticed. Note that the reference specimens were tested in air-dry condition at a mass moisture of 6%–8%. The effect of moisture on the value of flexural strength should be mainly ascribed to the weakening of the bonds between the crystals of the cement matrix structural lattice. The weakening is due to the fact that the bonds partially dissolve at a higher moisture content in the material, whereby the flexural strength (*MOR*) slightly decreases. As regards the path of the *σ*–*ε*, curve, one can see (Figure 4) that it clearly changed with the number of freeze–thaw cycles for both tested series of boards. Therefore it can be concluded that for the tested series of fibre-cement boards it was possible to determine the effect of the number of freeze–thaw cycles, as reflected in not only a reduction in flexural strength (*MOR*), but also in changes in the path of the *σ*–*ε* curve. It was found that when the number of cycles was increased from 1 to 10, the stiffness of the fibre-cement board and its brittleness decreased. In the case of the tested series, as the number of cycles increased, so did the range of the nonlinear increase in flexural stress while the limit of proportionality (*LOP*) considerably decreased. Thus, one can conclude that destructive changes took place in the structure of the fibre-cement boards. However, in the authors’ opinion, knowledge of the mechanical parameters is not enough to determine the damaging effect of freeze–thaw cycles on fibre-cement boards. Therefore, in order to better identify the changes taking place in the structure of the tested fibre-cement boards, the acoustic method and an artificial neural network were used in this research.

## 4. Investigations Conducted Using Acoustic Emission Method and Artificial Neural Network

The next step in this research on degrading changes in the structure of fibre-cement boards exposed to freeze–thaw cycling was an analysis of the AE signals registered in the course of the three-point bending test. The analysis was based on AE descriptors such as: events rate *N*_ev_, events sum ∑*N*_ev_ and events energy *E*_ev_, and on the signal frequency distribution. Figure 5 shows exemplary values of events sum ∑*N*_ev_ registered for the boards of series A_R_–A_10_ and B_R_–B_10_.

For a more precise analysis of the failure of the boards under bending and the effect of freeze–thaw cycling, events sum ∑*N*_ev_ and flexural stress *σ*_m_ versus time for selected cases are shown in Figure 6.

A clear fall in registered events and a change in the path of events sum ∑*N*_ev_ can be seen in Figure 6. The events were registered after the flexural strength (*MOR*) was exceeded. Note that a reduction in the number of events and a decrease in their energy were observed after subjecting the boards to freeze–thaw cycling. This does not mean that no AE events occurred, but one can suppose that the discrimination threshold for the registered events in the case of exposure to freeze–thaw cycling was too high. This can be connected with the measuring capability of the equipment used. Whereas the registered events with much lower energy could indicate a different process of destruction of the fibres—the latter can be pulled out of the wet cement matrix, whereby the event energy declines. These suppositions could be confirmed by SEM image analysis, which is presented further in this paper.

A spectral analysis of the AE event characteristics was carried out to more precisely identify the origins of the registered AE events. Reference acoustic spectra for cement matrix cracking, fibre breaking and the acoustic background were selected on this basis, which is described in detail in [12]. The reference acoustic spectra for cement matrix cracking were selected by analysing the record of acoustic activity in the time-frequency system during the bending of fibre boards previously fired in a laboratory furnace at a temperature of 230 °C for 3 h. It should be mentioned here that the investigations presented in this paper are part of a larger project devoted to the testing of fibre-cement boards. For example, in [11] it was observed that when fibre-cement boards were exposed to a temperature of 230 °C for 3 h, it resulted in the pyrolysis of their cellulose fibres, whereby the obtained fibre-cement board structure was completely devoid of fibres. Owing to this, reference acoustic spectra could be obtained for cement matrix cracking alone. The reference acoustic spectrum characteristic for fibre breaking was selected from the spectra obtained for air-dry reference boards of series A_R_ to B_R_, characterized by a repetitively similar characteristic in the frequency range of 10–24 kHz, clearly distinct from the cement matrix characteristic. The characteristic of the background acoustic spectrum, originating from the testing machine, was determined by averaging the characteristics for all the tested boards of series A and B in the initial phase of bending.

One should note that the selected spectral fibre breaking characteristics are understood as the signal accompanying the cracking of the cement matrix together with the fibres, whereas the spectral matrix characteristic is understood as the signal accompanying the cracking of the cement matrix alone. The selected acoustic spectrum characteristic reference standards were recorded every 0.5 kHz in 80 intervals. Figure 7 shows a record of the reference acoustic spectrum characteristics of the signal accompanying respectively cement matrix cracking and fibre breaking, and of the background.

Figure 7 shows that the background acoustic activity was at the level of 10–15 dB. The cement matrix acoustic spectrum characteristic reached the acoustic activity of 25 dB within frequency ranges 5–10 kHz (segment 1) and 20–32 kHz (segment 3). The acoustic activity of over 25 dB within frequency ranges 12–18 dB (segment 2) and 32–38 kHz (segment 4) was read off for the fibres.

The cement matrix, fibre and background reference standards were implemented in an artificial neural network, and the training and testing of the latter began. A unidirectional multilayer backpropagation structure with momentum was adopted for the ANN. A model of the artificial neuron is shown in Figure 8. The model includes *N* inputs, one output, a summation block and an activation block.

The following variables and parameters were used to describe the model shown in the Figure 8:
*x_i_* = (*x*_1_, *x*_2_, …, *x_N_*)   an input vector,(2)
*w_i_* = (*w*_1_, *w*_2_, …, *w_N_*)  a weight vector,(3)
*b* = −*θ* = *w*_0_  a bias,(4)
(5)$$v=u+b=\sum_{j=1}^{N}w_{j}x_{j}-\theta =\sum_{j=0}^{N}w_{j}^{}x_{j}\;a\;\mathrm{network}\;\mathrm{potential},$$
*F*(*v*)   an activation function.(6)

A model of the artificial neural network with inputs, information processing neurons and output neurons is shown in Figure 9.

Eight appropriate learning sequences were adopted iteratively to achieve optimal compatibility of the learned ANN with the training pattern, as presented in Table 3.

The spectral characteristics of fibre breaking were assigned to input A, the characteristics of cement matrix cracking were assigned to input B and the spectral characteristics of the background were assigned to input C. The spectral characteristics of fibre breaking were reproduced at input D.

After the ANN was trained on the input data, its mapping correctness was verified using the training and testing data. For this purpose, two pairs of input data were fed, that is, the data used for training the ANN, to check its ability to reproduce the reference spectra, and the one used for testing the ANN, to check its ability to identify the reference spectral characteristics originating from the fibres and the cement matrix during the bending test. For the eight performed training sequences, the ANN compliance with the training standard amounted to 0.995. Then, records of the ANN output in the form of recognised acoustic spectra for, respectively, fibre breaking, matrix cracking and the background were obtained. The learning coefficient (accelerating learning) was adopted at the level of 0.01 and momentum (increasing the stability of the obtained network configuration) at the level of 0.1. The following sigmoidal activation function was used: 1/(1 + exp(−*x*_i_)).

Figure 10 shows the results of the recognition of the reference acoustic spectrum standards. They are superimposed on the record of events rate *N*_ev_ and bending stress *σ*_m_ versus time. The diagrams are for the reference specimens of series A_R_ and for the series subjected to 1 and 10 freeze–thaw cycles (respectively A_1_ and A_10_). In order to better illustrate the recognized acoustic spectra, the matrix reference standards are marked green while the fibre reference standards are marked light brown.

Figure 10 clearly shows that the number of registered events for the fibre-cement boards subjected to 1 and 10 freeze–thaw cycles was lower in comparison with the reference boards. Slightly fewer events were registered after ten freeze–thaw cycles. These events had very low energy *E*_ev_ ranging from 10 to 100 nJ. An event originating from a cement matrix fracture initiates next events originating from the breaking of fibres or from their pulling out of the matrix. Under the influence of moisture and freeze–thaw cycles, the cement-fibre board became more plastic, which manifested in the disappearance of the interval in which strains were proportional to stresses. The matrix cracked once the flexural strength (*MOR*) was exceeded. The ten freeze–thaw cycles limited the registered events to solely cement matrix cracking after the exceedance of the flexural strength (*MOR*).

Table 4 shows events recognized as respectively accompanying fibre breaking and cement matrix cracking for the tested fibre-cement boards of series A_R_–A_10_.

Graphs of events rate *N*_ev_ and flexural stress *σ*_m_ versus time under freeze–thaw cycling, with superimposed identified reference spectral characteristics for the specimens of series B_R_, B_1_ and B_10_ are shown in Figure 11.

Figure 11 shows that the course of the AE signals registered for the fibre-cement boards of series B_1_ was similar to that for series B_R_, whereas the flexural stress *σ*_m_ curves differed. The limit of proportionality clearly fell and the interval in which the increment in stress relative to strain was nonlinear was wider. These changes can be ascribed to cement matrix yielding. The numerous events with the high energy of over 1000 nJ, originating from fibre breaking, indicate fibre destruction similar as in the reference boards. Therefore, it can be concluded that the fibres used in the tested boards did not change their properties under the influence of moisture and freeze–thaw cycling. This was confirmed by the high flexural strength (*MOR*) of about 27 MPa. In the case of the fibre-cement boards of series B_10_ subjected to ten freeze–thaw cycles, a slight decrease (about 20%) in flexural strength (*MOR*) and a drop in fibre breaking events in comparison with series B_1_ were observed. This means that the long-lasting dampness in conjunction with the freeze–thaw cycling had a degrading effect on the boards of series B_10_, reducing their strength and increasing their deformability. Moreover, the decrease in the sum of registered events in comparison with the reference boards could also be due to the lower acoustic activity of the events accompanying the breaking of the fibres, emitted below the discrimination threshold.

Summing up the results of the investigations, one can conclude that the effect of freeze–thaw cycling manifested itself mainly in a decrease in flexural strength (*MOR*) for all the tested series. The graphs presented in Figure 10 and Figure 11 show that when the flexural strength (*MOR*) was reached, the cement matrix cracked. The cracking of the cement matrix initiated events consisting of the breaking of the fibres. In all the series, the linear increment in stress relative to strain was found to clearly increase. The freeze–thaw cycling reduced the acoustic activity of the events taking place during the bending test. One can suppose that the different ways in which the fibres are destroyed as a result of freeze–thaw cycling can contribute to the decrease in the energy of fibre breaking events. Fibres can be damp and swollen and when undergoing destruction emit events characterized by very low energy. Whereas dampness-resistant PVA fibres (polyvinyl alcohol) instead of breaking can be pulled out of the damp cement matrix, which also will not generate high energy events. One can suppose that by optically examining the image of the fractured surface one can gain information about the mode of failure of the fibres.

Table 5 shows the events recognized as accompanying fibre breaking and cement matrix cracking for the boards of series B_R_–B_10_.

## 5. Investigations Conducted Using Scanning Electron Microscope (SEM)

A high-resolution environmental scanning electron microscope (SEM) Quanta 250 FEG, FEI with an EDS analyser was used for the investigations. Figure 12 and Figure 13 show exemplary images obtained by means of the SEM for respectively series A_R_–A_10_ and B_R_–B_10_. The examined fibre-cement boards had previously been subjected to the three-point bending test.

On the basis of an analysis of the SEM and EDS images, the macrostructure of the fibre-cement boards of series A_R_ and B_R_ can be described as compact. The microscopic examinations revealed the structure to be fine-pore, with pore size of up to 50 µm. Cavities up to 500 µm wide and grooves were visible in the places in the fractured surface where fibres had been pulled out. In the images one can see cellulose and PVA fibres. Various forms of hydrated calcium silicates of the C-S-H type occurred, with an “amorphous” phase and a phase comprised of strongly adhering particles predominating. An analysis of the composition of the fibres showed elements native to them and elements native to cement. The surface of the fibres was coated with a thin layer made up of cement matrix and hydration products. The fact that there were few places with a space between the fibres and the matrix indicates that they were strongly bonded. An examination of the fibre-cement boards subjected to one or ten freeze–thaw cycles reveals that most of the fibres were pulled out of the cement matrix. Numerous cavities, most left by the pulled-out cellulose fibres, and grooves were visible. The matrix structure was more granular and included numerous delaminations. Strongly compact structures having an irregular shape were found to be present.

Summing up the results of the investigations, the authors conclude that the freeze–thaw cycling of fibre-cement boards affected their structure. Fibres wholly pulled out of the matrix and the voids left by them in the cement matrix were visible in the fibre-cement boards subjected to freeze–thaw cycling. This was especially noticeable in the fibre-cement boards subjected to 10 freeze–thaw cycles, mainly in the boards of series A. In the case of the reference boards, much more fibres (whose ends were firmly “anchored” in the cement matrix) were ruptured. This conclusively proves that freeze–thaw cycling significantly weakened the structure of fibre-cement boards.

## 6. Conclusions

The impact of low temperatures, in the form of freeze–thaw cycling, is by nature destructive to most building products. The degree of resistance to this impact is measured by the number of freeze–thaw cycles after which such products retain the properties specified by the standards. The investigations of the fibre-cement boards of series A and B carried out as part of this study showed that the boards differed in their degree of resistance to the impact of freeze–thaw cycling. Thanks to the use of the acoustic emission method during the three-point bending of the boards, the course of their failure for one or ten freeze–thaw cycles could be observed and compared with that of the reference boards. An artificial neural network was employed to analyse the results yielded by the acoustic emission method. The investigations conclusively proved that freeze–thaw cycling affected the way fibre-cement boards failed, as reflected mainly in the decrease in the number of AE events recognized as accompanying the breaking of fibres during the three-point bending of the specimens. SEM examinations were carried out in order to obtain better insight into the changes taking place in the structure of the tested boards. They confirmed that significant changes took place in the structure of the boards, especially after 10 freeze–thaw cycles. The structure became less compact (more granular). Most of the fibres were not ruptured during the bending test, but pulled out of the cement matrix, as confirmed by the low energy of the registered AE events and their small number. This was particularly evident in the case of the tested fibre-cement boards of series A. In the authors’ opinion, the above findings are important for building practice since they indicate that it is inadvisable to use fibre-cement boards whose flexural strength (*MOR*) considerably decreases under the influence of freeze–thaw cycling for the cladding of ventilated façades, especially in high buildings located in zones of high wind load.

## Figures and Tables

**Figure 1 materials-12-02181-f001:**
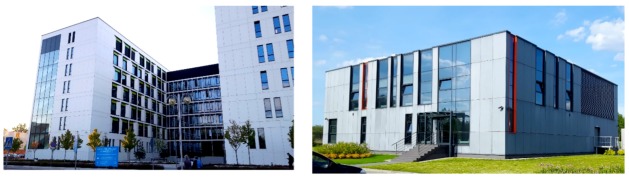
Exemplary uses of fibre-cement boards as ventilated façade cladding.

**Figure 2 materials-12-02181-f002:**
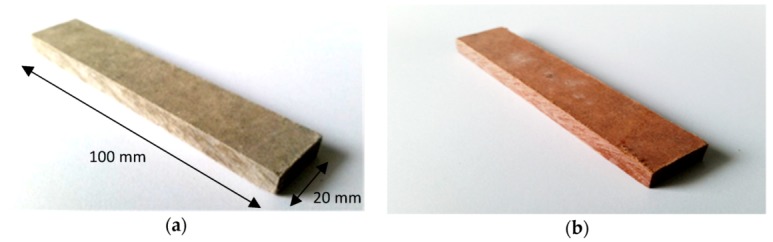
Tested fibre-cement board specimens: (**a**) board A, (**b**) board B.

**Figure 3 materials-12-02181-f003:**
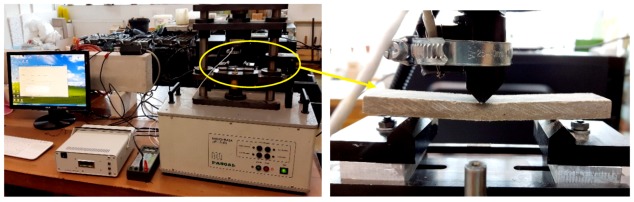
(**left**) Three-point bending test stand and acoustic emission (AE) measuring equipment and (**right**) close-up of fibre-cement board specimen during test.

**Figure 4 materials-12-02181-f004:**
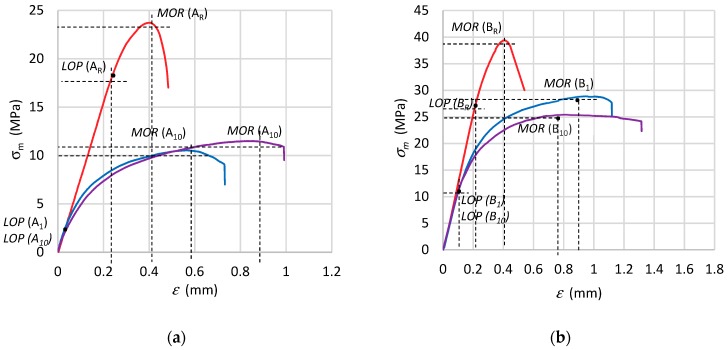
Bending σ–ε relation for specimens of fibre-cement boards: (**a**) series A, (**b**) series B. *LOP*: limit of proportionality; *MOR*: modulus of rupture.

**Figure 5 materials-12-02181-f005:**
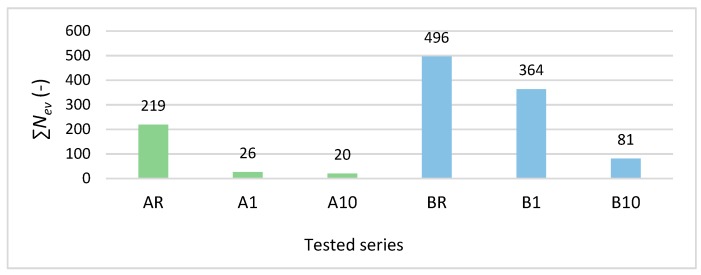
Exemplary events sum ∑*N*_ev_ values registered for air-dry specimens and specimens subjected to freeze–thaw cycling.

**Figure 6 materials-12-02181-f006:**
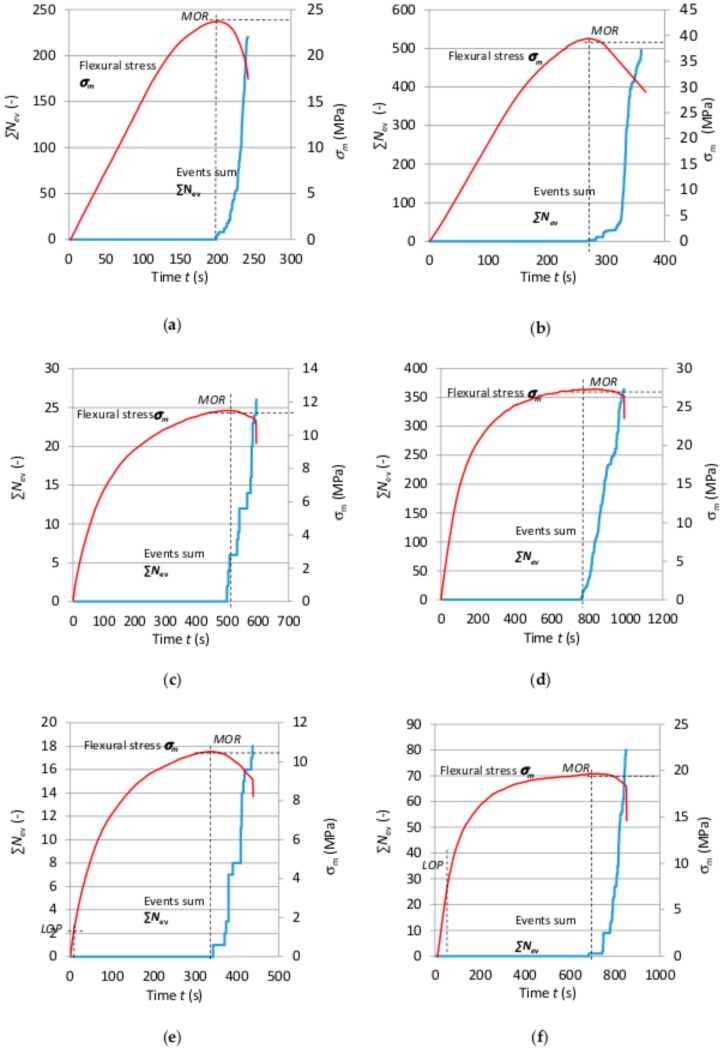
Flexural stress *σ*_m_ and events sum ∑*N*_ev_ as function of time *t* for boards of: (**a**) series A_R_, (**b**) series B_R_, (**c**) series A_1_, (**d**) series B_1_, (**e**) series A_10_, (**f**) series B_10_.

**Figure 7 materials-12-02181-f007:**
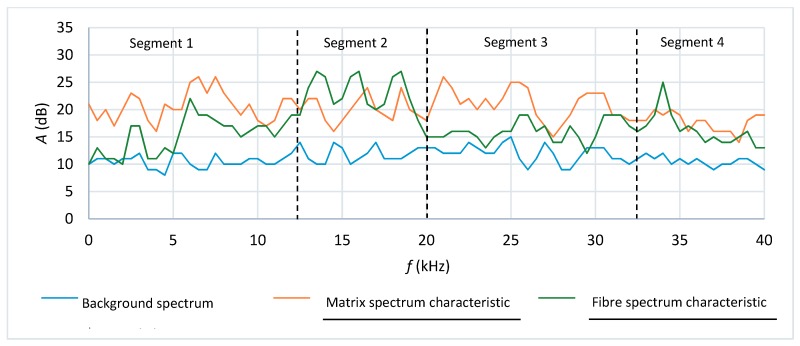
Background, fibre and cement matrix acoustic spectrum characteristics as function of frequency [12].

**Figure 8 materials-12-02181-f008:**
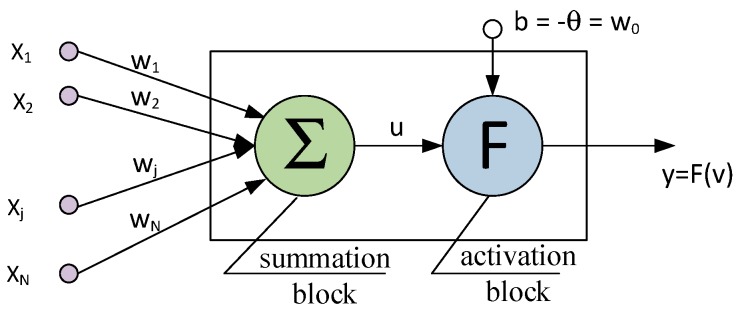
Model of an artificial neuron [13,42].

**Figure 9 materials-12-02181-f009:**
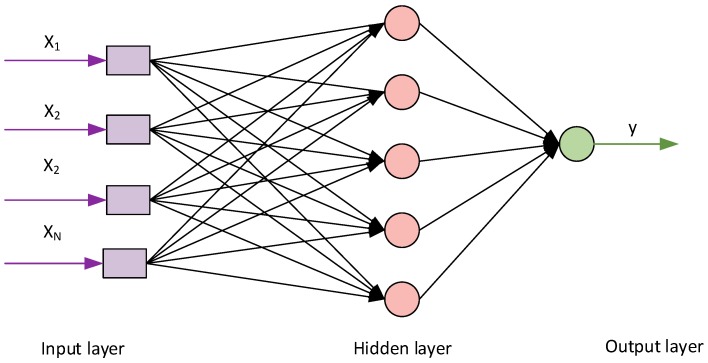
Model of the artificial neural network [13,43].

**Figure 10 materials-12-02181-f010:**
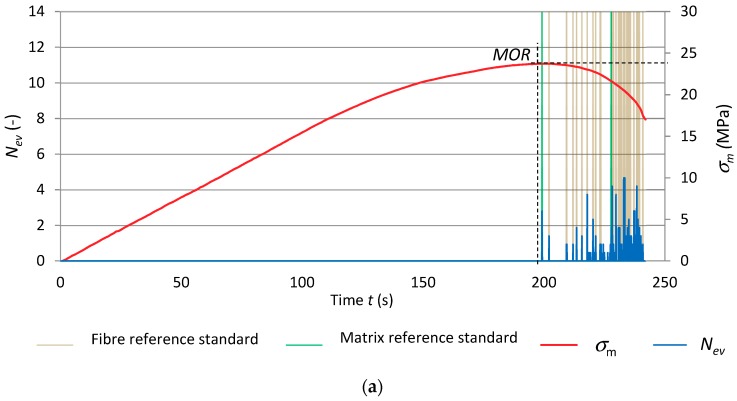
Freeze–thaw cycling diagrams of events rate *N*_ev_ and bending stress σ_m_ versus time, with superimposed identified reference spectral characteristics: (**a**) series A_R_, (**b**) series A_1_, (**c**) series A_10_.

**Figure 11 materials-12-02181-f011:**
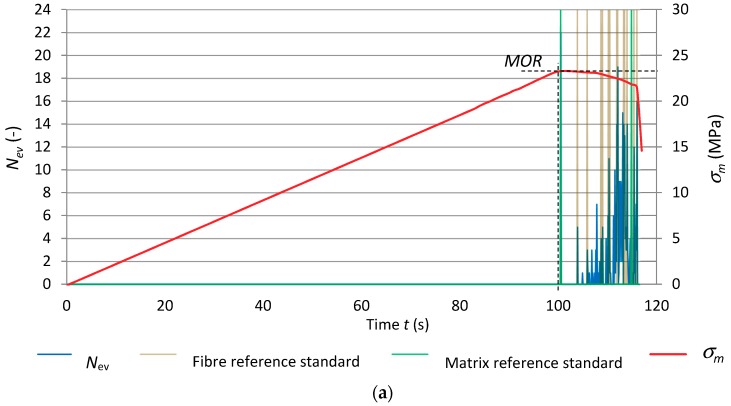
Events rate *N*_ev_ and flexural stress σ_m_ versus time, under freeze–thaw cycling, with superimposed identified reference spectral characteristics for: (**a**) series B_R_, (**b**) series B_1_, (**c**) series B_10_.

**Figure 12 materials-12-02181-f012:**
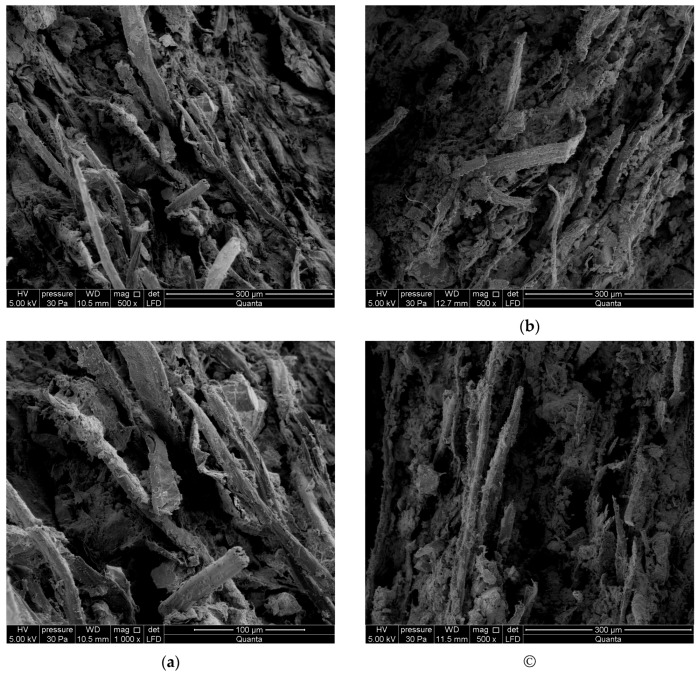
SEM images of boards: (**a**) series A_R_ (magnification 1000× at top, 500× at bottom), (**b**) series A_1_, (**c**) series A_10_.

**Figure 13 materials-12-02181-f013:**
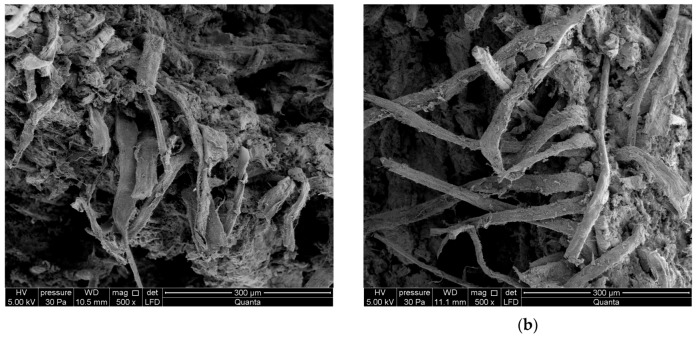
SEM images of boards: (**a**) series B_R_ (magnification 1000× at top, 500× at bottom), (**b**) series B_1_, (**c**) series B_10_.

**Table 1 materials-12-02181-t001:** Basic specifications of the tested fibre-cement boards of series A and B.

Series	Board Thickness *e* (mm)	Board Colour	Application	Board Bulk Density *ρ* (g/cm^3^)	Flexural Strength *MOR* (MPa)
A	8.0	natural	exterior	1.65	24
B	8.0	full body coloured	exterior	1.58	38

**Table 2 materials-12-02181-t002:** Series of fibre-cement boards and test cases and their denotations.

Series Name/Test Case	Series A	Series B
Air-dry condition (reference board)	A_R_	B_R_
1 freeze–thaw cycles	A_1_	B_1_
10 freeze–thaw cycles	A_10_	B_10_

**Table 3 materials-12-02181-t003:** ANN learning sequences.

No.	Learning Sequences	No.	Learning Sequences
1	16,000 × A, 8000 × B, 4000 × C, 2000 × D	5	16,000 × A, 8000 × B, 4000 × C
2	16,000 × A, 8000 × B, 4000 × C, 2000 × D	6	16,000 × A, 8000 × B, 4000 × C
3	16,000 × A, 8000 × B, 4000 × C, 2000 × D	7	16,000 × A, 8000 × B, 1000 × C
4	16,000 × A, 8000 × B, 4000 × C	8	16,000 × A, 8000 × B

**Table 4 materials-12-02181-t004:** Events recognized as respectively accompanying fibre breaking and cement matrix cracking for fibre-cement boards of series A_R_–A_10_.

Series	Events Sum	Sum of Recognized Events	Sum of Events Assigned to Fibre Breaking	Sum of Events Assigned to Matrix Cracking
	∑*N*_ev_	∑*N*_ev,r_	∑*N*_ev,f_	∑*N*_ev,m_
A_R_	219	201	193	8
A_1_	26	24	20	4
A_10_	20	19	16	3

**Table 5 materials-12-02181-t005:** Events recognized as accompanying fibre breaking and cement-matrix cracking for boards of series B_R_–B_10_.

Series	Events Sum	Sum of Recognized Events	Sum of Events Assigned to Fibre Breaking	Sum of Events Assigned to Matrix Cracking
	∑*N*_ev_	∑*N*_ev,r_	∑*N*_ev,f_	∑*N*_ev,m_
B_R_	496	487	439	48
B_1_	364	350	332	18
B_10_	81	76	68	8

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
