# Peer review of "Effect of Freeze–Thaw Cycling on the Failure of Fibre-Cement Boards, Assessed Using Acoustic Emission Method and Artificial Neural Network"

_materials, 2019, doi:10.3390/ma12132181_

Round 1

Reviewer 1 Report

The authors present a work on the effect of freeze-thaw cycling on 12 the failure of fibre-cement boards using acoustic emission and artificial neural network techniques. The subject of the authors work is an important significant issue in structural engineering and materials, and such an attempt is of great interest. 

However the paper, in its present form, requires some substantial modifications in order to justify its publication in an International Journal such as Materials. The following points should be further elaborated by the authors:

Despite the fact that the use of ANNs is the main subject of the articles, authors have not supported this subct. Namely, authors are kindly requested 

a) to present in depth and detailed the database used in their work. Furthermore, it will be usuful to present it as supplementary materials,

b) to describe in depth the methodology used for the training and the development of the ANN models (number of input and output parameters, number of hidden layers, training algorithm, activation functions,

c) the performance criteria used for the evaluation of the ANN mathematical models.

Also, 

The literature review is not adequately covered and complete as presented in the manuscript. Please consider add some more references. One recommended:

Asteris, P.G., Kolovos, K.G. (2019). Self-compacting concrete strength
prediction using surrogate models, Neural Computing and Applications,
31, 409-424.

Author Response

We are deeply grateful to the Reviewer for the effort put in the review of our paper.

We agree with most of the Reviewer’s comments and we have taken them into account in the paper’s revised version.

The authors are convinced that many of the Reviewer’s suggestions will be helpful in further research and analyses which will form the basis for the next paper.

a)      Some details of the database used in work as much as we can was presented and marked by yellow colour in the text.

b)      The methodology used for the training and the development of the ANN models (number of input and output parameters, number of hidden layers, training algorithm, activation functions was presented and marked by yellow colour in the text.

c)      The performance criteria used for the evaluation of the ANN mathematical models was presented and marked by yellow colour in the text.

Suggested reference was added:

Asteris, P.G., Kolovos, K.G. (2019). Self-compacting concrete strength prediction using surrogate models, Neural Computing and Applications, 31, 409-424.

Once again we would like to thank the Reviewer most warmly for the perceptive and detailed comments, which greatly enhance the understanding of the paper and its value.

Reviewer 2 Report

Title:  Effect of freeze-thaw cycling on failure of fibre-cement boards, assessed using acoustic emission method and artificial neural network  

This paper investigate the effect of freeze-thaw cycling on the failure of fibre-cement boards and the changes taking place in their structure.  The research work indicates that freeze-thaw cycling has an effect on the failure of fibre-cement boards. The topic is interesting and fits well with the scope of the journal.  

1 Literature review of the manuscript is not enough and more latest paper should be included. There are a lot of published papers in term of studying the free-thaw cycling on the behavior of fiber-cement boards, which is not new and this topic has been investigated widely in the previous studies.  So, it is not clear what makes the present study unique. What is the gap of literature, what is the main contribution of the study, which is different from others should be clearly clarified? It needs to clearly state the contributions of the manuscript in the introduction section. Although the study conducted many experimental tests, the experimental results did not make the manuscript novelty.

2 There are a lot of previous study on the application of fiber-cement board and the interface in civil engineering.  A precise and comprehensive literature review upon the application in various perspectives should be well evaluated. Some of the related literatures are recommended for authors to refer as follows: Sensors and Actuators A: Physical, 2009, 153: 166-170; Cement and Concrete Composites, 2017, 80: 287-297; Composites Part B: Engineering, 2016, 90: 392-398.

3 The title and abstract needs to be revised to scientific describe the research work. Meanwhile, the important results from experiments or the main contribution of the manuscript should be summarized.

4 The manuscript needs to be thoroughly edited for English text. The English is deficient in several parts (e.g. Introduction, Investigations conducted using acoustic emission method and artificial neural network).

5 In this study, a lot of experimental results, artificial neural network and SEM have been presented. How can the findings from current research work benefit the research and practical application of fiber-cement board? Please evaluate more recommendations or insights in this direction.

Author Response

We are deeply grateful to the Reviewer for the effort put in the review of our paper.

We agree with most of the Reviewer’s comments and we have taken them into account in the paper’s revised version.

The authors are convinced that many of the Reviewer’s suggestions will be helpful in further research and analyses which will form the basis for the next paper.

1 Literature review of the manuscript is not enough and more latest paper should be included. There are a lot of published papers in term of studying the free-thaw cycling on the behavior of fiber-cement boards, which is not new and this topic has been investigated widely in the previous studies.  So, it is not clear what makes the present study unique. What is the gap of literature, what is the main contribution of the study, which is different from others should be clearly clarified? It needs to clearly state the contributions of the manuscript in the introduction section. Although the study conducted many experimental tests, the experimental results did not make the manuscript novelty.

We generally agree with Reviewer that there are some published papers in term of studying the free-thaw cycling on the behavior of fiber-cement boards. The problem is that almost all of them described mechanical parameters e.g. MOR, unfortunately not described changes in fiber cement boards structures. In our opinion this is the gap of literature and the main contribution of the study, which is different from others (line: 46-50, 60-65 in the text).

2 There are a lot of previous study on the application of fiber-cement board and the interface in civil engineering.  A precise and comprehensive literature review upon the application in various perspectives should be well evaluated. Some of the related literatures are recommended for authors to refer as follows: Sensors and Actuators A: Physical, 2009, 153: 166-170; Cement and Concrete Composites, 2017, 80: 287-297; Composites Part B: Engineering, 2016, 90: 392-398.

We generally agree with Reviewer that there are a lot of very interesting published papers in term of studying the free-thaw cycling on the behavior of concrete, unfortunately not described too much fibre-cement boards.

3 The title and abstract needs to be revised to scientific describe the research work. Meanwhile, the important results from experiments or the main contribution of the manuscript should be summarized.

This has been complied with as much as we can.

4 The manuscript needs to be thoroughly edited for English text. The English is deficient in several parts (e.g. Introduction, Investigations conducted using acoustic emission method and artificial neural network).

The linguistic errors have been corrected. The paper has been checked by a sworn translator of the English language.

5 In this study, a lot of experimental results, artificial neural network and SEM have been presented. How can the findings from current research work benefit the research and practical application of fiber-cement board? Please evaluate more recommendations or insights in this direction.

In the authors’ opinion, the above findings are important for building practice since they indicate that it is inadvisable to use fibre-cement boards whose flexural strength (MOR) considerably decreases under the influence of freeze-thaw cycling for the cladding of ventilated façades, especially in high buildings located in zones of high wind load.

The authors are convinced that many of the Reviewer’s suggestions will be helpful in further research and analyses which will form the basis for the next paper.

Once again we would like to thank the Reviewer most warmly for the perceptive and detailed comments, which greatly enhance the understanding of the paper and its value.

Reviewer 3 Report

The paper is about a study of the effect of freeze-thaw cycling on failure of fibre cement boards. Because it is supported on laboratory tests, it is always interesting for any reader, so the subject of the paper is interesting for publishing.

However, in my opinion the paper is not suitable for publishing in the present version.

There are 38 references in the text, being about 60% from the last 5 years, and about 21% are more than 10 years old, which is totally acceptable.

In line 38, instead of “figure” it should be “Figure”. The same applies to lines 196, 197, 275 and 302.

There is no Figure 5, so figures should be renumbered.

In line 133, instead of “table” it should be “Table”. The same applies to line 143.

After Equation (1), in line 161, it should be explained the meaning of “b” and “Ls”.

My main concerns about the paper are related to the use of an artificial neural network (ANN) to obtain the results.

In line 59, a reference should be added about ANN, because there are many readers that don’t have any knowledge about this mater, and because no detailed explanation is presented in the paper. For example, the following paper:

M. LEFIK  (2013) - Some aspects of application of artificial neural network for numerical modeling in civil engineering. BULLETIN OF THE POLISH ACADEMY OF SCIENCES TECHNICAL SCIENCES, Vol. 61, No. 1. DOI: 10.2478/bpasts-2013-0003

It is stated, very briefly, that a “unidirectional multilayer backpropagation structure with momentum was adopted for the ANN”. However, it is not well explained how many layers and neurons were adopted. Also, it is stated that “input training and testing data were fed into the ANN”, but no detailed description of the training data is presented in the paper.

How can the authors be sure about the results of the ANN?

The use of an ANN is an interesting approach to obtain correlations between input variables and output results, working mainly as a black box. However, the capabilities of the ANN are much dependent on the architecture of the ANN and on the characteristics of the training set, which can influence the accuracy of the results. For example, see the influence of the adopted training data in the results in the following paper:

Estêvão, João M. C. (2018). Feasibility of using neural networks to obtain simplified capacity curves for seismic assessment. Buildings, 8(11), 151. doi: 10.3390/buildings8110151

An ANN is not a “magic box”, that solves all problems. It is a very powerful tool, indeed, but the accuracy of the results is much, much dependant on the ANN architecture and how it was trained, otherwise the results might not be reliable.

This issue should be mentioned in the paper, and a reference added, and the authors should clarify how they have solved this problem, presenting a detailed description about the neural network that was adopted in the present study (input variables, output variables, number of layers and neurons), training parameters, training data (a detailed description), accuracy tests (mean errors and maximum errors), and software used.

See, for example:

Seung-Chang Lee (2003) - Prediction of concrete strength using artificial neural networks. Engineering Structures. 25 849–857. Doi: 10.1016/S0141-0296(03)00004-X

For all those reasons, I believe that the paper is only acceptable for publishing after a major revision.

Author Response

We are deeply grateful to the Reviewer for the effort put in the review of our paper.

We agree with most of the Reviewer’s comments and we have taken them into account in the paper’s revised version.

The authors are convinced that many of the Reviewer’s suggestions will be helpful in further research and analyses which will form the basis for the next paper.

In line 38, instead of “figure” it should be “Figure”. The same applies to lines 196, 197, 275 and 302.

This has been complied with and marked by yellow colour in the text.

There is no Figure 5, so figures should be renumbered.

This has been complied with and marked by yellow colour in the text.

In line 133, instead of “table” it should be “Table”. The same applies to line 143.

This has been complied with and marked by yellow colour in the text.

After Equation (1), in line 161, it should be explained the meaning of “b” and “Ls”.

This has been complied with and marked by yellow colour in the text.

In line 59, a reference should be added about ANN, because there are many readers that don’t have any knowledge about this mater, and because no detailed explanation is presented in the paper. For example, the following paper:

M. LEFIK  (2013) - Some aspects of application of artificial neural network for numerical modeling in civil engineering. BULLETIN OF THE POLISH ACADEMY OF SCIENCES TECHNICAL SCIENCES, Vol. 61, No. 1. DOI: 10.2478/bpasts-2013-0003

Estêvão, João M. C. (2018). Feasibility of using neural networks to obtain simplified capacity curves for seismic assessment. Buildings, 8(11), 151. doi: 10.3390/buildings8110151

Seung-Chang Lee (2003) - Prediction of concrete strength using artificial neural networks. Engineering Structures. 25 849–857. Doi: 10.1016/S0141-0296(03)00004-X

Suggested reference was added. Also more details and explanation about ANN was added and marked by yellow colour in the text.

Once again we would like to thank the Reviewer most warmly for the perceptive and detailed comments, which greatly enhance the understanding of the paper and its value.

Round 2

Reviewer 1 Report

-

Author Response

Once again we would like to thank the Reviewer most warmly for the perceptive and detailed comments, which greatly enhance the understanding of the paper and its value.

Reviewer 2 Report

From the revised version, there is not many revisions and authors seem to deny and ignore the reviewers' comments and suggestions. Please carefully consider all comments and suggestions from all Reviewers.

Author Response

Once again we are deeply grateful to the Reviewer for the effort put in the review of our paper.

We agree with most of the Reviewer’s comments and we have taken them into account in the paper’s revised version.

Please believe and take into consideration that we do not deny and not ignore the Reviewers' comments and suggestions. I am so sorry if you have filled up like this. This was not our goal.

What is important we carefully consider all comments and suggestions from all Reviewers and we tried to find the best solution.

Follow to that we corrected our paper trying to include the comments of all Reviewers, although some of them were mutually exclusive.

Once again the linguistic errors have been corrected and once again the paper has been checked by a sworn translator of the English language. This person is well known English translator and working for the government office. We never had a problem with his translation, but any way we asked him to do that once again.

Once again we would like to thank the Reviewer most warmly for the perceptive and detailed comments, which greatly enhance the understanding of the paper and its value.

Sincerely,

Prof. Krzysztof Schabowicz

Tomasz Gorzelańczyk

Reviewer 3 Report

The authors have improved the original paper and have addressed most of my concerns.

However, there are still some minor corrections to carry out.

In line 165, where is “B” it should be “b”.

The paper is acceptable for publishing after minor revision.

Author Response

We are deeply grateful to the Reviewer for the effort put in the review of our paper.

We agree with most of the Reviewer’s comments and we have taken them into account in the paper’s revised version.